# Structures of Toxic Advanced Glycation End-Products Derived from Glyceraldehyde, A Sugar Metabolite

**DOI:** 10.3390/biom14020202

**Published:** 2024-02-08

**Authors:** Akiko Sakai-Sakasai, Kenji Takeda, Hirokazu Suzuki, Masayoshi Takeuchi

**Affiliations:** 1Department of Advanced Medicine, Medical Research Institute, Kanazawa Medical University, 1-1 Daigaku, Uchinada, Kahoku 920-0293, Ishikawa, Japan; asakasai@kanazawa-med.ac.jp (A.S.-S.); ktakeda@kanazawa-med.ac.jp (K.T.); 2General Medicine Center, Kanazawa Medical University Hospital, 1-1 Daigaku, Uchinada, Kahoku 920-0293, Ishikawa, Japan; 3Department of Cardiology, Kanazawa Medical University, 1-1 Daigaku, Uchinada, Kahoku 920-0293, Ishikawa, Japan; 4Department of Organic and Medicinal Chemistry, Faculty of Pharmaceutical Sciences, Hokuriku University, Kanazawa 920-1181, Ishikawa, Japan; h-suzuki@hokuriku-u.ac.jp

**Keywords:** advanced glycation end-products (AGEs), glyceraldehyde (GA), GA-derived AGEs (GA-AGEs), toxic AGEs (TAGE), lifestyle-related diseases (LSRDs)

## Abstract

Advanced glycation end-products (AGEs) have recently been implicated in the onset/progression of lifestyle-related diseases (LSRDs); therefore, the suppression of AGE-induced effects may be used in both the prevention and treatment of these diseases. Various AGEs are produced by different biological pathways in the body. Glyceraldehyde (GA) is an intermediate of glucose and fructose metabolism, and GA-derived AGEs (GA-AGEs), cytotoxic compounds that accumulate and induce damage in mammalian cells, contribute to the onset/progression of LSRDs. The following GA-AGE structures have been detected to date: triosidines, GA-derived pyridinium compounds, GA-derived pyrrolopyridinium lysine dimers, methylglyoxal-derived hydroimidazolone 1, and argpyrimidine. GA-AGEs are a key contributor to the formation of toxic AGEs (TAGE) in many cells. The extracellular leakage of TAGE affects the surrounding cells via interactions with the receptor for AGEs. Elevated serum levels of TAGE, which trigger different types of cell damage, may be used as a novel biomarker for the prevention and early diagnosis of LSRDs as well as in evaluations of treatment efficacy. This review provides an overview of the structures of GA-AGEs.

## 1. Introduction

Under hyperglycemic conditions, glucose (Glu) reacts with primary amines (the N-terminal or lysine (Lys) side chain) or the guanidine group of the arginine (Arg) side chain of proteins. This process begins with the conversion of reversible Schiff base adducts to more stable, covalently bound Amadori rearrangement products. Over the course of several days to weeks, these Amadori products undergo further dehydration, condensation, fragmentation, rearrangement, and oxidation reactions to form irreversibly bound moieties known as advanced glycation end-products (AGEs) [1,2,3,4]. AGEs exogenously or endogenously accumulate in the body [5,6,7]. Various AGEs are produced in these reactions, with the different types formed being dependent on the reducing sugars and carbonyl compounds involved [8,9,10,11,12,13,14,15,16,17]. The rate of glycation also differs according to the types of sugars involved, with glyceraldehyde (GA), a tricarbohydrate and intermediate of Glu and fructose (Fru) metabolism, being faster in glycation reactions than Glu/Fru [18,19,20,21]. GA-derived AGEs (GA-AGEs) are highly cytotoxic [22,23].

In our recent studies, we showed that intracellularly accumulated GA-AGEs contributed to the onset/progression of lifestyle-related diseases (LSRDs) by triggering cell damage [24,25,26,27,28,29,30,31,32,33,34,35,36]. Elevated intracellular levels of GA-AGEs have also been implicated in the pathogenesis of metabolic syndrome (MetS), non-alcoholic steatohepatitis (NASH), cardiovascular diseases (CVD), diabetes mellitus (DM) and its complications, some cancer types, and Alzheimer’s disease (AD) [22,23,37,38,39]. Although the mechanisms by which GA-AGEs exert their toxic effects and the specific structures responsible have not yet been elucidated, previous studies have demonstrated that an antibody targeting GA-AGEs can effectively attenuate AGE-induced neurotoxicity in the serum of patients with diabetic nephropathy with hemodialysis (DN-HD) [40]. However, this protective effect was not observed when antibodies targeting other types of AGEs were used. Therefore, AGE structures containing epitopes recognized by the anti-GA-AGE antibody solely appear to be toxic, are called toxic AGEs (TAGE), and are distinct from other GA-AGEs, including triosidines [41], GA-derived pyridinium compounds (GLAP) [42], pyrrolopyridinium Lys dimers derived from GA (PPGs) [43], methylglyoxal (MGO)-derived hydroimidazolone 1 (MG-H1) [44], and argpyrimidine (ArgP) [45].

This review provides an overview of the structures involved in the formation of AGEs, particularly for GA-AGEs.

## 2. Routes by Which Various AGEs Are Produced in the Human Body

We previously demonstrated that AGEs were produced not only from reducing sugars, such as Glu, Fru, and GA, but also from carbonyl compounds, including glycolaldehyde (Glycol), glyoxal (GO), MGO, and 3-deoxyglucosone [12,14,16]. We also developed AGE antibodies that specifically recognize seven distinct AGE classes among circulating proteins and peptides in serum collected from DN-HD. Based on our findings, we proposed pathways for the in vivo formation of distinct AGE classes by glycation, as shown in Figure 1.

Marked variations have been reported in the structures of AGEs in vivo and their production is known to be dependent on a number of complex reactions, including dehydration, oxidation, condensation, and rearrangements; therefore, their abundant formation is expected. However, the structures of only approximately 30 types of AGEs have been elucidated to date [46,47,48,49,50,51], and those of highly cytotoxic AGEs currently remain unknown. These AGEs are classified into four groups according to their chemical characteristics:Fluorescent and cross-linked products (i.e., PPGs, pentosidine, crossline, and vespelysine);Fluorescent and non-cross-linked products (i.e., ArgP);Non-fluorescent and cross-linked products (i.e., MGO-Lys-dimer, GO-Lys-dimer, and glucosepane);Non-fluorescent and non-cross-linked products (i.e., trihydroxy-triosidine, GLAP, MG-H1, N^ε^-(carboxyethyl)Lys, N^ε^-(carboxymethyl)Lys (CML), and pyrraline).

## 3. Generation of GA

GA is a very potent glycating agent that is produced by the pathways responsible for sugar metabolism, including fructolysis, glycolysis, and the polyol pathway [38], and plays an important role in the generation of GA-AGEs (Figure 2).

### 3.1. Fructolysis

Fru is a common sugar in diets and is the main component of high-Fru corn syrup. It is metabolized in cells via two pathways: (1) Phosphorylation by hexokinase, which is present in all cells. However, hexokinase exhibits a strong preference for Glu, which is present at a concentration of approximately 5 mM in blood and, thus, is a strong competitive inhibitor of Fru phosphorylation. (2) Fructokinase (FK) in the liver, particularly after a meal. Fru is phosphorylated by a specific FK in the liver to form Fru-1-phosphate (F-1-P), which is then cleaved by aldolase B [52,53]. The products of Fru metabolism in the liver are dihydroxyacetone-phosphate (DHA-P) and GA.

### 3.2. Glycolysis

Glycolysis is a fundamental metabolic pathway for the conversion of Glu to pyruvate. GA-3-phosphate (GA-3-P) is a key intermediate of this pathway and a substrate of GA-3-P dehydrogenase (GAPDH). Together with a reduction in GAPDH activity, the metabolism of GA-3-P decreases and GA-3-P accumulates intracellularly. GA-3-P metabolism shifts to another route and the amount of GA increases, which promotes the formation of GA-AGEs. As shown in Figure 2, GA-3-P is an intermediate of this pathway, undergoes enzymatic reduction by GAPDH, and eventually forms pyruvate. Some GA-3-P is also spontaneously degraded to GA, which reacts non-enzymatically with proteins to create GA-AGEs. GA-AGEs have been implicated in diseases related to hyperglycemia [54,55].

### 3.3. Polyol Pathway

In healthy individuals, Glu is converted to GA-3-P by GAPDH and is maintained at low levels [56]. Aldose reductase (AR) catalyzes the reduction of Glu to sorbitol, which is then converted to Fru by sorbitol dehydrogenase. AR has a low affinity for Glu and, thus, this pathway is not very active under normal Glu homeostasis. However, under hyperglycemic conditions, Glu concentrations are elevated in insulin-independent tissues, such as red blood cells, the brain and nerve tissues, the lens, and the kidneys, which enhances the activity of the polyol pathway [57,58]. Fru may also be metabolized to GA through fructolysis by FK and aldolase B.

Therefore, changes in the regulation of sugar metabolic pathways may induce an increase in GA levels and, thus, the accumulation of GA-AGEs, resulting in cytotoxicity [23].

## 4. Generation of MGO

MGO is a highly reactive dicarbonyl compound that is formed as a byproduct of glycolysis, in which the intermediate triosephosphates, DHA-P and GA-3-P, spontaneously degrade to generate MGO (Figure 2) [59]. Approximately 0.05–0.1% of the triosephosphates in glycolysis are degraded to MGO [60]. MGO may contribute to the production of AGEs through selective reactions with the Arg and Lys side chains on proteins. MGO is an Arg-directed glycating agent that mainly forms MG-H1 [61,62] and ArgP [63,64]. MGO is metabolized by multiple pathways to prevent its accumulation to abnormally high levels. The glyoxalase (GLO) system, comprising GLO 1 and 2, is the primary metabolic pathway for MGO, converting it to D-lactate [65,66,67,68].

## 5. Structures of GA-AGEs

To date, the following GA-AGE structures have been detected: triosidines [41], GLAP [42], PPGs [43], MG-H1 [44], and ArgP [45] (Figure 1 and Figure 2). Trihydroxy-triosidine, GLAP, and PPGs were identified in the Lys residues and MG-H1 and ArgP in the Arg residues of proteins. Even though the cytotoxicity of GA-AGEs has been examined, few studies have attempted to identify GA-AGEs using a proteome-wide analysis. Senavirathna et al. reported that Lys residues were significantly more likely to form GA-AGEs than Arg residues, which will be important for the future development of novel strategies to prevent and treat GA-AGE-dependent diseases [69]. This finding also has important implications for the prevention and treatment of LSRDs.

### 5.1. Structures of GA-AGEs with Lys Modifications

The roles of the highly reactive triose sugar GA in protein cross-linking and Lys/Arg modifications during the Maillard reaction have been investigated. However, the structurally distinct GA-AGEs involved in LSRDs currently remain unknown. The modification of proteins by GA may generate a number of different GA-AGEs, including trihydroxy-triosidine, GLAP, PPGs, MG-H1, and Arg P (Figure 3).

#### 5.1.1. Triosidines

Tessier et al. incubated GA with N^α^-acetyl-L-Lys and N^α^-acetyl-L-Arg and isolated four new Maillard reaction pyridinium compounds named triosidines [41]. Lys adducts are trihydroxy-triosidine, Lys-hydroxy-triosidine, and triosidine-carbaldehyde, the latter of which is labile. The Lys-Arg adduct is Arg-hydroxy-triosidine. Triosidines are a novel class of AGEs that are expected to offer novel insights of biomedical importance for disorders of triose metabolism and the biophysical and toxic effects of triose-based diseases.

#### 5.1.2. GLAP

GLAP is formed from GA with N^α^-acetyl-L-Lys [42]. Since GLAP is not formed from MGO, it has potential as a specific marker of reduced GAPDH activity in metabolic diseases, such as diabetic complications. Although GLAP, which is formed by GA-related glycation, was identified in the plasma proteins and tail tendon collagen of streptozotocin-induced diabetic rats [70], it remains unclear whether circulating GLAP levels may be used as a biomarker of vascular injury in diabetic patients. Usui et al. revealed that GLAP derived from GA induced oxidative stress in HL-60 cells [71]. GLAP also exerted toxic effects in PC12 neuronal cells, which were inhibited by an anti-receptor for AGEs (RAGE) antibody [72]. Matsui et al. demonstrated that GLAP interacted with RAGE and induced the production of reactive oxygen species (ROS) in endothelial cells (ECs) [73]. However, the precise structural moieties required for the toxic effects of GLAP via RAGE have not yet been fully elucidated.

#### 5.1.3. PPGs

The pyridinium-type AGEs trihydroxy-triosidine and GLAP have been identified as GA-AGEs; however, their pathological effects are yet to be clarified. The Maillard reaction between N^α^-acetyl-L-Lys and GA was recently examined under physiological conditions and the main products identified were PPG1 and PPG2, which are fluorescent AGEs in which pyrrolopyridine ring structures cross-link two Lys molecules [43]. Based on these findings, a mechanism was proposed for the generation of PPG1 and PPG2 in vitro via a unique reaction to expand the skeleton as an alternative to the typical route for the formation of Schiff bases.

### 5.2. Structures of GA-AGEs with Arg Modifications

Although the majority of AGEs are formed from the Lys/Arg residues of proteins, the structures of GA-AGEs produced from Arg residues remain unclear.

#### 5.2.1. MG-H1

Usui et al. isolated and identified GA-AGEs formed from GA and N^α^-acetyl-L-Arg. A major product was identified as MG-H1 [44]. They isolated and identified MG-H1, which was generated in the presence of Lys/Arg residues, and suggested that MG-H1 was formed through both GA-related and MGO-related pathways. MG-H1 has potential as an indicator of damage to GA- and MGO-related enzymes, such as GAPDH and GLO. Rabbani et al. recently reported a relationship between serum MG-H1 levels and the stage of chronic kidney disease [74]. Zhang et al. also showed that plasma MG-H1 levels were associated with impaired cognitive performance in Chinese community-dwelling middle-aged and elderly subjects [75]. However, further studies are needed to clarify the relationship between blood MG-H1 levels and LSRDs.

#### 5.2.2. ArgP

Three major GA-AGEs were formed from a mixture of N^α^-acetyl-L-Lys, N^α^-acetyl-L-Arg, and GA. Two of these compounds were GLAP and MG-H1, as previously reported, while the other was identified as ArgP. Although ArgP is a modified product of Arg residues, it did not form from GA with Arg residues [45]. The co-existence of Lys residues is necessary for the formation of ArgP, which is considered to be derived from MGO; however, a number of pathways are involved in the formation of ArgP. ArgP may be generated by both MGO and GA and, thus, has potential as a marker of both MGO-/GA-related glycation.

## 6. Structures of TAGE

GA is one of the most potent precursors of GA-AGEs because it causes the non-enzymatic glycation of proteins, resulting in irreversible TAGE production. The formation of TAGE mainly occurs intracellularly through glycation reactions between GA and intracellular proteins, which produce TAGE molecules of different sizes and properties. Although a more detailed understanding of the mechanisms underlying TAGE toxicity is needed, previous studies have indicated that it is due to oxidative stress-induced damage [26,39], the loss of protein functions as a result of glycation modifications [39], and TAGE aggregation and accumulation [30].

### 6.1. Proposed TAGE Structures

TAGE were prepared by a model system, involving a reaction between egg lysozyme and GA, and samples were separated by sodium dodecyl sulfate polyacrylamide gel electrophoresis. After their migration, the separated proteins were transferred to PVDF membranes and an anti-TAGE antibody recognized dimer/trimer bands (intermolecular protein cross-links) and monomer bands (intramolecular protein cross-links) [unpublished data]. Two compounds with GA-derived 1,4-dihydropyrazine rings emitting AGE-specific fluorescence and forming intra/intermolecular cross-links have been proposed as candidate TAGE structures (Figure 4) [76].

### 6.2. New Concept in LSRDs: The TAGE Theory

Although GA-AGEs are a structurally heterogeneous group of molecules, the specific structures responsible for their toxicity have yet to be identified.

The neurotoxicity of AGEs in the serum of DN-HD was previously shown to be completely suppressed by an anti-TAGE antibody, and this protective effect was not achieved when antibodies targeting other types of AGEs were used [40]. Therefore, solely AGE structures containing epitopes recognized by the anti-GA-AGE antibody appear to be toxic. Specific GA-AGEs recognized by the anti-TAGE antibody are called TAGE and are distinct from other AGEs and GA-AGEs, such as triosidines, GLAP, PPGs, MG-H1, and ArgP. The accumulation of intracellular GA-AGEs has been implicated in the onset/progression of LSRDs. This discovery has led to the TAGE concept.

## 7. Cytotoxicity of TAGE

Previous studies have suggested the involvement of TAGE accumulation in the pathogenesis of NASH, CVD, DM and its complications, some cancer types, and AD [22,23,37,38,39,55,77]. We recently demonstrated that TAGE were produced by and accumulated in many cell types [24,25,26,27,28,29,30,31,32,33,34,35,36]. Excessive TAGE damage cells, leak into surrounding cells and the circulation, interact with RAGE, and the activation of the TAGE–RAGE axis contributes to the onset/progression of LSRDs [54,55]. We previously reported the binding affinity of the human RAGE protein to TAGE, which were more abundant in the serum of patients with DM than in healthy controls. The dissociation constant of TAGE with RAGE was 0.36 μM [78]. In DM, TAGE have been suggested to exert intracellular effects by strongly binding to RAGE.

### 7.1. Accumulation of Intracellular TAGE and Cell Damage

#### 7.1.1. Intracellular TAGE and Cell Death

Intracellular GA levels are increased by the frequent and excessive intake of sugar-sweetened beverages and/or processed foods, which are prevalent in contemporary daily diets. Reactions between elevated levels of GA and intracellular proteins induce the production of TAGE [79], which exert cytotoxic effects in various cell types, including hepatocytes [24,25,26,27], neuroblastoma cells [28,29,30], cardiomyocytes [31] and cardiac fibroblasts (CFs) [32], pancreatic ductal cells [33], β-cells [34], myoblasts [35], and osteoblasts [36]. Based on these findings, TAGE have been suggested to play a role in apoptosis and/or necrosis, which ultimately lead to cell death and tissue damage.

##### TAGE and Hepatocytes

Cell damage caused by the production and accumulation of TAGE has been examined using molecular biological methods with established human hepatic parenchymal cell lines (Hep3B and HepG2) and primary cultured hepatocytes. The findings obtained revealed the following: (i) when GA was added to hepatocytes, the cell viability decreased as the TAGE levels increased [24,25,27]; (ii) the molecular chaperone Hsc70 [24] and the apoptosis execution factor caspase-3 [25] were identified as proteins that undergo TAGE modifications; (iii) ROS production was associated with the intracellular accumulation of TAGE, suggesting a mitochondrial abnormality as the cause [26]; (iv) the mRNA levels of C-reactive protein, a marker of inflammation, significantly increased after the addition of GA [24]. These findings indicate that the production and accumulation of intracellular TAGE induce cell damage and inflammatory reactions. Therefore, strategies to suppress the intracellular accumulation of TAGE have potential for the prevention and treatment of the onset/progression of NASH [39].

##### TAGE and Neurons

A GA-treated human neuroblastoma (SH-SY5Y) cell extract was obtained by two-dimensional electrophoresis, detected using the TAGE-specific antibody, and target proteins were identified using a time-of-flight mass spectrometer. One of the target proteins was β-tubulin, which plays an important role in the formation of microtubules in nerve axons. The GA-induced formation of TAGE on β-tubulin may result in the generation of TAGE–β-tubulin aggregates, thereby inhibiting nerve axon elongation [29]. The phosphorylation of tau, which is present in neurofibrillary tangles, was also observed. The formation of TAGE–β-tubulin was found to induce similar pathological changes to neurofibrillary tangles in AD, which contribute to the onset/progression of AD. Abnormal aggregation associated with the production of TAGE–β-tubulin appeared to inhibit the normal formation of microtubules, thereby suppressing neuronal axon elongation. Furthermore, pyridoxamine (PM), a type of vitamin B6, attenuated the inhibition of nerve axon elongation by suppressing the formation of TAGE [30]. Therefore, a more detailed understanding of the production of TAGE–β-tubulin and the mitigating effects of PM will provide insights into the mechanisms underlying the onset of AD.

##### TAGE, Cardiomyocytes, and CFs

We previously reported that the intracellular accumulation of TAGE reduced the beat rate of cardiomyocytes and this was accompanied by a decrease in autophagy function and cell death [31]. On the other hand, extracellular TAGE did not directly inhibit cardiomyocyte beating or induce cardiomyocyte cell death. In our recent study, we immunologically identified Hsp90β and its high-molecular-weight (HMW) complex as one of the proteins that undergo TAGE modifications in cardiomyocytes. We also showed that the protein levels of LC3-II and the LC3-II/LC3-I ratio decreased with TAGE modifications to Hsp90β [unpublished data]. Based on these findings, TAGE affect the functions of proteins that regulate autophagy-related pathways, such as those involved in the production of LC3-II. Furthermore, when CFs were treated with GA, cell death was induced due to the production and accumulation of intracellular TAGE [32]. However, the addition of TAGE in excess of a physiological concentration to CFs did not induce cell death. Therefore, we speculate that the intracellular accumulation of TAGE induced cell death in CFs and directly suppressed cardiac repair.

#### 7.1.2. Mechanisms Underlying Intracellular TAGE Degradation

Intracellularly accumulated TAGE have been implicated in the onset and progression of diseases. The trigger for TAGE accumulation is an elevation in intracellular GA levels. However, it remains unclear whether there is an endogenous defense system that recognizes and eliminates TAGE as aberrant intracellular proteins. The findings of our recent study, which utilized mutated forms of checkpoint kinase 1 (CHK1), a serine-threonine kinase central to cell cycle checkpoints and the DNA damage response (DDR) [80], strongly suggest the existence of a cellular pathway that recognizes and degrades TAGE [81].

CHK1, glycation-modified by a GA stimulation, formed HMW complexes that gradually accumulated intracellularly. The loss of CHK1 function due to TAGE modifications may induce cell death through DNA damage. Furthermore, CHK1 underwent the cleavage of its C-terminal regulatory domain by SPRTN metalloprotease, forming CHK1 cleavage products (CHK1-CPs) comprising the constitutively activated N-terminal kinase domain. In contrast to the accumulation of intact CHK1 with glycation, CHK1-CPs decreased in a time-dependent manner in response to the GA stimulation, indicating that GA not only modified CHK1 through glycation, leading to functional impairment, but also contributed to the degradation of highly active CHK1-CPs, thereby inhibiting the DDR pathway. It is important to note that in response to the GA stimulation, CHK1-CPs and their mimic (d270WT) were rapidly degraded primarily through the ubiquitin–proteasome pathway. Furthermore, an inactive mutant variant of d270WT underwent glycation and ubiquitin modifications more rapidly than d270WT when stimulated with GA. On the other hand, p62, which is involved in selective autophagy, appeared to be one of the components of these HMW complexes that escaped degradation [81].

The recognition of structural changes induced by glycation and their ubiquitination suggest the presence of specific ubiquitin ligases. Future research needs to focus not only on the relationship between TAGE formation and diseases, but also on identifying the ubiquitin ligases and degradation pathways triggered by structural changes due to glycation. Advances in this field will contribute to the development of effective interventions for diseases related to protein aggregation and AGE accumulation, such as DM, AD, and CVD.

#### 7.1.3. Extracellular TAGE and the TAGE–RAGE Axis

A great variety of precursors and complex mechanisms of AGE formation generate many chemically diverse AGE molecules that have so far been identified in foods and in human blood/tissues [82,83,84,85,86]. Most often, processed foods in modern diets are rich in sugars and proteins that, during thermal processing, undergo the Maillard reaction, leading to rapid AGE formation [87,88,89,90]. CML, one of the most well-described AGEs, was used as a representative for the determination of AGE levels in food products [91].

Glu/Fru metabolic pathways in cells produce GA, which, in turn, generates intracellular TAGE and, ultimately, results in cell death. The extracellular leakage of TAGE affects the surrounding cells via interactions with RAGE. TAGE accumulation and the up-regulated expression of RAGE are promoted by the frequent and excessive intake of dietary AGEs [92], resulting in TAGE–RAGE interactions, which produce ROS. ROS have been shown to up-regulate RAGE expression and increase the production of TAGE, which have been suggested to play a role in the onset/progression of LSRDs [93,94,95,96].

Moreover, the interactions between TAGE and RAGE have been shown to elicit oxidative, inflammatory, and thrombogenic responses in ECs [96,97]. We recently focused on RAS guanyl nucleotide-releasing protein 2 (RASGRP2), a guanine nucleotide exchange factor that activates small GTPases. We showed that RASGRP2 expressed in human vascular ECs activated RAP1 and R-RAS, which suppressed apoptosis. Furthermore, using the established RASGRP2 stable overexpression line, we conducted an analysis with a focus on TAGE-induced vascular endothelial dysfunction, particularly barrier dysfunction. The findings obtained revealed that the localization of VE-cadherin and ZO-1 was disrupted and vascular permeability increased. In addition, RASGRP2 suppressed TAGE-induced vascular permeability by preventing the disturbance of VE-cadherin localization through the activation of RAP1 and R-RAS signals. We suggested that TAGE increased the vascular permeability by disrupting both adherens and tight junctions through intricate ROS and non-ROS signaling pathways [98].

Yamagishi’s group recently reported that RAGE-aptamer significantly inhibited the binding of AGEs, senescent macroprotein derivatives formed at an accelerated rate under DM, to RAGE, and, as a result, the onset/progression of diabetic nephropathy, melanoma growth and metastasis, and renal injury were attenuated in animal models of chronic kidney disease [99,100,101,102,103].

These findings indicate the therapeutic potential of targeting the TAGE–RAGE axis to prevent vascular complications in DM.

### 7.2. Involvement of Circulating TAGE in LSRDs

Intracellular TAGE accumulation has been implicated in the pathogenesis of a number of cellular disorders, and extracellular leakage elevates the circulating TAGE levels [23,37]. We previously reported increases in the serum TAGE levels under oxidative stress and inflammatory/hyperglycemic conditions and showed that they were associated with insulin resistance, endothelial dysfunction, vascular inflammation, and a decreased number and reduced migratory activity of endothelial progenitor cells (EPCs) in humans [23].

Collectively, our findings indicate the potential of serum TAGE levels as a biomarker for the prevention and early diagnosis of LSRDs as well as in evaluations of treatment efficacy [23]. (i) Increased serum TAGE levels were observed not only in DM patients, but also in non-DM states, and (ii) the serum TAGE levels were significantly higher in patients with NASH, which is a liver phenotype of MetS, than in healthy subjects and the non-alcoholic fatty liver group. (iii) Even in healthy subjects, a decrease in the number of vascular EPCs was observed in the group with high serum TAGE levels, indicating the potential of TAGE as markers for predicting the progression of arteriosclerosis in the future. (iv) In healthy subjects administered collagen tripeptide, which inhibits TAGE, reductions were observed in the serum TAGE levels and the cardio-ankle vascular index also decreased. In other words, reductions in TAGE levels may restore the elasticity of blood vessels and prevent arteriosclerosis. (v) Examinations of the relationship between serum TAGE levels and continued pregnancy rates in patients being treated for infertility revealed that the continued pregnancy rate was low in groups with high serum TAGE levels, even at younger ages. Following the administration of water chestnut extract to infertility-refractory patients and another round of infertility treatment, the serum TAGE levels were reduced and the birth rate increased. (vi) We also conducted an observational study on the relationship between serum TAGE levels before the diagnosis of colorectal cancer (CRC) and mortality, and found that the serum TAGE levels correlated with CRC-specific mortality and all-cause mortality. A strong relationship was observed between TAGE levels and CRC-specific mortality in patients with distal colon cancer [104]. Therefore, these findings support a direct relationship between prediagnostic serum TAGE levels in CRC patients and CRC-specific and all-cause mortality.

Elevated serum TAGE levels may be used in predictions of the onset/progression of LSRDs, even in healthy subjects with normal blood test results; therefore, the similar application of circulating TAGE levels is possible. In future studies, high-risk patients may be identified by measuring their TAGE levels and important information may be obtained that will accelerate treatment decision making. A more detailed understanding of serum TAGE levels will facilitate early predictions of the onset/progression of disease. However, there is no supportive evidence for a relationship between the blood GA-AGE levels and diseases using anti-triosidine, anti-GLAP, and anti-PPG antibodies. Based on the findings discussed in this review, specific GA-AGE structures in TAGE may play roles in numerous types of cell injury both in vitro and in vivo.

## 8. Conclusions and Perspectives

Based on research performed by our group, we proposed the TAGE theory [22,23,37,38,39]. Increases in the serum TAGE levels as a result of different types of cell damage may be used as a novel biomarker for the prevention and early diagnosis of LSRDs as well as in evaluations of treatment efficacy, including DM and non-DM [23]. Early predictions of the onset/progression of LSRDs may be achieved based on measurements of serum TAGE levels, which will contribute to the prevention of LSRDs and extend the healthy life expectancy.

Among the many categories of AGEs, GA-AGEs are denoted as TAGE due to their inherent toxicity. Notably, anti-TAGE antibodies did not recognize known GA-AGE structures containing pyridine rings, such as triosidines, GLAP, and PPGs. In our proposed framework, two compounds featuring GA-derived 1,4-dihydropyridine rings have emerged as candidates for the structures of TAGE when they undergo dimerization and trimerization, respectively [76]. The anti-TAGE antibody identified dimeric/trimeric bands (representing intermolecular protein cross-links) and monomeric bands (indicating intramolecular protein cross-links) in the reaction mixture of a lysozyme protein and GA. In consideration of the diverse array of proteins within cells, each with varying molecular sizes and subcellular localizations, it is conceivable that higher-order cross-linked TAGE structures may also be present. Consequently, a more exhaustive and intricate structural analysis of TAGE is imperative for future investigations.

The introduction of the TAGE theory presents a novel research perspective for LSRDs.

## Figures and Tables

**Figure 1 biomolecules-14-00202-f001:**
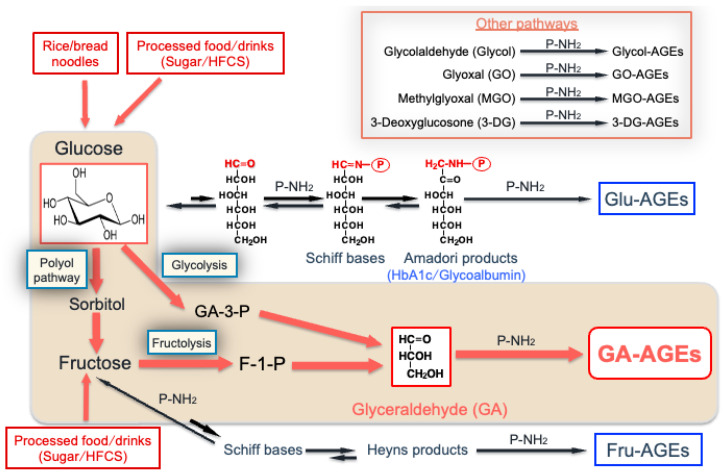
Routes by which advanced glycation end-products (AGEs) are produced in the human body. HFCS: high-fructose corn syrup; HbA1c: hemoglobin A1c; Glu-AGEs: glucose-derived AGEs; GA-AGEs: glyceraldehyde-derived AGEs; Fru-AGEs: fructose-derived AGEs; Glycol-AGEs: glycolaldehyde-derived AGEs; GO-AGEs: glyoxal-derived AGEs; MGO-AGEs: methylglyoxal-derived AGEs; 3-DG-AGEs: 3-deoxyglucosone-derived AGEs; GA: glyceraldehyde; P-NH_2_: free amino residue of a protein.

**Figure 2 biomolecules-14-00202-f002:**
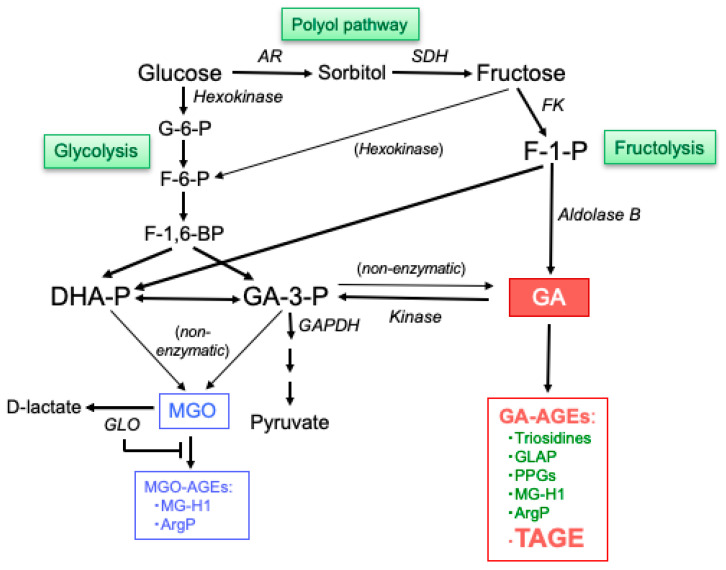
Routes by which GA and MGO are generated through sugar metabolic pathways. G-6-P: glucose-6-phosphate; F-6-P: fructose-6-phosphate; F-1,6-BP: fructose-1,6-bisphosphate; F-1-P: fructose-1-phosphate; AR: aldose reductase; SDH: sorbitol dehydrogenase; FK: fructokinase; GAPDH: GA-3-phosphate dehydrogenase; GLO: glyoxalase; GLAP: GA-derived pyridinium compounds; PPGs: pyrrolopyridinium Lys dimers derived from GA; MG-H1: MGO-derived hydroimidazolone 1; ArgP: argpyrimidine; TAGE: toxic AGEs.

**Figure 3 biomolecules-14-00202-f003:**
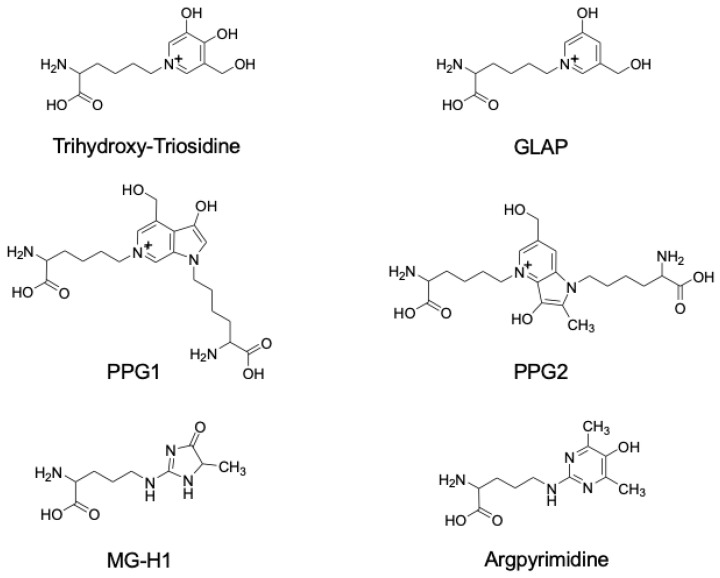
Structures of glyceraldehyde (GA)-derived AGEs. GLAP: GA-derived pyridinium; PPG: pyrrolopyridinium Lys dimer derived from GA; MG-H1: MGO-derived hydroimidazolone 1.

**Figure 4 biomolecules-14-00202-f004:**
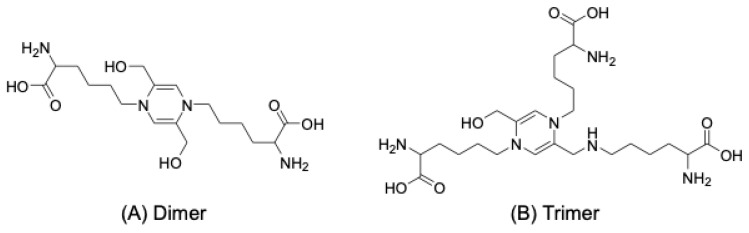
Proposed structures of TAGE. (**A**): 1,4-Di(5-amino-5-carboxypentyl)-2,5-dihydroxymethyl-1,4-dihydropyrazine; (**B**): 1,4-di(5-amino-5-carboxypentyl)-5-(5-amino-5-carboxy-pentylaminomethyl)-2-hydroxymethyl-1,4-dihydropyrazine.

## Data Availability

Not applicable.

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
