# Peer review of "Structures of Toxic Advanced Glycation End-Products Derived from Glyceraldehyde, A Sugar Metabolite"

_biomolecules, 2024, doi:10.3390/biom14020202_

Round 1
Reviewer 1 Report
Comments and Suggestions for Authors
In this review, the authors provide an overview of the structures of advanced glycation end-products derived from glyceraldehyde, an intermediate of glucose and fructose metabolism.
The work is well organised and comprehensively described. References are appropriate and adequate.
In general, it is a significant contribution in the area of proteins and/or peptide glycation.
Two minor recommendations:
Page 3, line 107. Gylcolysis should be substituted by glycolysis.
Figure 3. The resolution of Figure 3 should be improved. Some chemical structures containing a positively charged nitrogen atom are not well distinguished: trihydroxy-triosidine, PPG1, PPG2 and GLAP.
Author Response
We would like to thank the Reviewers for their careful consideration of our manuscript. The comments provided were very helpful and we hope that our responses are satisfactory. Point-by-point responses to each Reviewer are shown in red.
1) Page 3, line 107. Gylcolysis should be substituted by glycolysis.
(A): We have corrected it according to your instructions.
2) Figure 3. The resolution of Figure 3 should be improved. Some chemical structures containing a positively charged nitrogen atom are not well distinguished: trihydroxy-triosidine, PPG1, PPG2 and GLAP.
(A): We increased the resolution of Figure 3 according to your suggestion.
Reviewer 2 Report
Comments and Suggestions for Authors
This is an interesting review of Advanced Glycation End-Products derived from glyceraldehyde (AGEs), especially toxic AGEs (here named "TAGEs").
A commendable point is that the structures of these compounds are presented, and this is very useful and important. On the other hand, Figure 5 seems a bit exaggerated to me, as it is mainly based on hypotheses, and should be removed (or modified, to include only the effects already known to be caused by AGEs).
Another point: 38 out of 95 citations are self-citations. Perhaps it is a bit of an exaggeration in the self-citations since this subject is widely studied by different groups around the world.
Author Response
We would like to thank the Reviewers for their careful consideration of our manuscript. The comments provided were very helpful and we hope that our responses are satisfactory. Point-by-point responses to each Reviewer are shown in red.
1) On the other hand, Figure 5 seems a bit exaggerated to me, as it is mainly based on hypotheses, and should be removed (or modified, to include only the effects already known to be caused by AGEs).
(A): As you pointed out, we removed Figure 5.
2) Another point: 38 out of 95 citations are self-citations. Perhaps it is a bit of an exaggeration in the self-citations since this subject is widely studied by different groups around the world.
(A): Following your suggestion, we reduced the ratio of self-cited papers to about 1/3.